# Structure-Aware Image Segmentation with Homotopy Warping

**Xiaoling Hu**

Department of Computer Science
Stony Brook University
xiaolhu@cs.stonybrook.edu

## Abstract

Besides per-pixel accuracy, topological correctness is also crucial for the segmentation of images with fine-scale structures, e.g., satellite images and biomedical images. In this paper, by leveraging the theory of digital topology, we identify pixels in an image that are critical for topology. By focusing on these critical pixels, we propose a new **homotopy warping loss** to train deep image segmentation networks for better topological accuracy. To efficiently identify these topologically critical pixels, we propose a new algorithm exploiting the distance transform. The proposed algorithm, as well as the loss function, naturally generalize to different topological structures in both 2D and 3D settings. The proposed loss function helps deep nets achieve better performance in terms of topology-aware metrics, outperforming state-of-the-art structure/topology-aware segmentation methods.

## 1 Introduction

Image segmentation with topological correctness is a challenging problem, especially for images with fine-scale structures, e.g., satellite images, neuron images and vessel images. Deep learning methods have delivered strong performance in image segmentation task [33, 22, 7, 8, 9]. However, even with satisfying per-pixel accuracy, most existing methods are still prone to topological errors, i.e., broken connections, holes in 2D membranes, missing connected components, etc. These errors may significantly impact downstream tasks. For example, the reconstructed road maps from satellite images can be used for navigation [3, 5]. A small amount of pixel errors will result in broken connections, causing incorrect navigation route. See Fig. 1 for an illustration. In neuron reconstruction [18, 27, 49, 54, 53], incorrect topology of the neuron membrane will result in erroneous merge or split of neurons, and thus errors in morphology and connectivity analysis of neuronal circuits.

Topological errors usually happen at challenging locations, e.g., weak connections or blurred locations. But not all challenging locations are topologically relevant; for example, pixels at the boundary of the object of interest can generally be challenging, but not relevant to topology. To truly suppress topological errors, we need to focus on *topologically critical pixels*, i.e., challenging locations that are topologically relevant. Without identifying and targeting these locations, neural networks that are optimized for standard pixel-wise losses (e.g., cross-entropy loss or mean-square-error loss) cannot avoid topological errors, even if we increase the training set size.

Existing works have targeted these topologically critical pixels. The closest method to our work is TopoNet [24], which is based on persistent homology [15, 14]. The main idea is to identify topologically critical pixels corresponding to critical points of the likelihood map predicted by the neural network. The selected critical points are reweighed in the training loss to force the neural network to focus on them, and thus to avoid topological errors. But there are two main issues with this approach: 1) the method is based on the likelihood map, which can be noisy with a large amount

of irrelevant critical points. This leads to inefficient optimization during training. 2) The computation for persistent homology is cubic to the image size. It is too expensive to recompute at every iteration.

In this paper, we propose a novel approach to identify topologically critical pixels in a more efficient and accurate manner. These locations are penalized in the proposed *homotopy warping loss* to achieve better topological accuracy. Our method is partially inspired by the warping error previously proposed to evaluate the topological accuracy [26]. Given a binary predicted mask $f_B$ and a ground truth mask $g$, we "warp" one towards another without changing its topology. From the view of topology, to warp mask $f_B$ towards $g$, we find a mask $f_B^*$ that is homotopy equivalent to $f_B$ and is as close to $g$ as possible [21]. The difference between the warped mask $f_B^*$ and $g$ constitutes the topo-

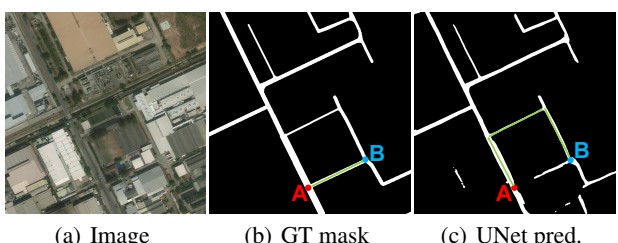

(a) Image      (b) GT mask      (c) UNet pred.

Figure 1: An illustration for the importance of topological correctness. If one wants to go to point $B$ from $A$, the shortest path in the GT is illustrated in **(b)** (the green path). However, in the result predicted by UNet, though only a few pixels are misclassified, the shortest path from $A$ to $B$ is totally different, which is illustrated by the green path in **(c)**. Zoom-in for better viewing.

logically critical pixels. We can also warp $g$ towards $f_B$ and find another set of topologically critical pixels. See Fig. 4 and Fig. 5 for illustrations. These locations directly correspond to topological difference between the prediction and the ground truth, tolerating geometric deformations. Our homotopy warping loss targets them to fix topological errors of the model.

The warping of a mask is achieved by iteratively flipping labels at pixels without changing the topology of the mask. These flippable pixels are called *simple points/pixels* in the classic theory of digital topology [30]. Note that in this paper, we focus on the topology of binary masks, simple points and simple pixels can be used interchangeably. To find the optimal warping of a mask towards another mask is challenging due to the huge search space. To this end, we propose a new heuristic method that is computationally efficient. We filter the image domain with the distance transform and flip simple pixels based on their distance from the mask. This algorithm is proven efficient and delivers high quality locally optimal warping results.

Overall, our contributions can be summarized as follows:

- We propose a novel *homotopy warping loss*, which penalizes errors on topologically critical pixels. These locations are defined by homotopic warping of predicted and ground truth masks. The loss can be incorporated into the training of topology-preserving deep segmentation networks.

- By exploiting distance transforms of binary masks, we propose a novel homotopic warping algorithm to identify topologically critical pixels in an efficient manner. This is essential in incorporating the homotopy warping loss into the training of deep nets.

Our loss is a plug-and-play loss function. It can be used to train any segmentation network to achieve better performance in terms of topology. We conduct experiments on both 2D and 3D benchmarks to demonstrate the efficacy of the proposed method. Our method performs strongly in multiple topology-relevant metrics (e.g., ARI, Warping Error and Betti Error). We also conduct several ablation studies to further demonstrate the efficiency and effectiveness of the technical contributions.

## 2 Related Works

**Deep Image Segmentation.** Deep learning methods (CNNs) have achieved satisfying performances for image segmentation [33, 7, 8, 9, 36, 39]. By replacing fully connected layer with fully convolutional layers, FCN [33] transforms a classification CNNs (e.g., AlexNet [31], VGG [43], or ResNet [23]) to fully-convolutional neural networks. In this way, FCN successfully transfers the success of image classification [31, 43, 47] to dense prediction/image segmentation. Instead of using Conditional Random Field (CRF) as post-processing, Deeplab (v1-v2) [7, 8] methods add another fully connected CRF after the last CNN layer to make use of global information. Moreover, Deeplab

v3 [9] introduces dilated/atrous convolution to increase the receptive field and make better use of context information to achieve better performance.

Besides the methods mentioned above, UNet [39] has also been one of the most popular methods for image segmentation, especially for images with fine structures. UNet architecture is based on FCN with two major modifications: 1) Similar to encoder-decoder, UNet is symmetric. The output is with the same size as input images, thus suitable for dense prediction/image segmentation, and 2) Skip connections between downsampling and upsampling paths. The skip connections of UNet are able to combine low level/local information with high level/global information, resulting in better segmentation performance.

Though obtaining satisfying pixel performances, these methods are still prone to structural/topological errors, as they are usually optimized via pixel-wise loss functions, such as mean-square-error loss (MSE) and cross-entropy loss. As illustrated in Fig.1, a small amount of pixel errors will affect or even damage the downstream tasks.

**Topology-Aware Segmentation.** Topology-aware segmentation methods have been proposed to segment with correct structure/topology. By identifying critical points of the predicted likelihood maps, persistent-homology-based losses [24, 11] penalize topologically critical pixels. However, the identified critical points can be very noisy and often are not relevant to the topological errors. Illustrations are included in Sec. A.1. Moreover, the computation of persistent homology is expensive, making it difficult to evaluate the loss and gradient at every training iteration.

Other methods indirectly preserve topology by enhancing the curvilinear structures. VGG-UNet [35] uses the response of pretrained filters to enhance structures locally. But it does not truly preserve the topology, and cannot generalize to higher dimensional topological structures, such as voids. Several methods extract skeletons of the masks and penalize heavily on pixels of the skeletons. This ensures the prediction to be correct along the skeletons, and thus are likely correct in topology. clDice [42] extracts the skeleton through min/max-pooling operations over the likelihood map. DMT Loss [25] uses the Morse complex of the likelihood map as the skeleton. However, these skeletons are not necessarily topologically critical. The penalization on them may not be relevant to topology.

We also note that many deep learning techniques have been proposed to ensure the segmentation output preserves details, and thus preserves topology implicitly [39, 33, 2, 13, 29, 28]. One may also use topological constraints as postprocessing steps once we have the predicted likelihood maps [20, 32, 46, 41, 51, 19, 50, 37, 55, 6, 1, 45, 38, 16]. Compared to end-to-end methods, postprocessing methods usually contain self-defined parameters or hand-crafted features, making it difficult to generalize to different situations.

Instead of relying on the noisy likelihood maps [24, 11, 42, 25], we propose to use the warping of binary masks to identify the topologically critical pixels. The identified locations are more likely to be relevant to topological errors. Penalizing on these locations ensures the training efficiency and segmentation quality of our method. Another difference from previous methods is that our method rely on purely local topological computation (i.e., checking whether a pixel is simple within a local patch), whereas previous methods are mostly relying on global topological computation.

## 3 Method

By warping the binary predicted mask to the ground truth or conversely, we can accurately and efficiently identify the critical pixels. And then we propose a novel homotopy warping loss which targets them to fix topological errors of the model. The overall framework is illustrated in Fig. 2.

This section is organized as follows. We will start with necessary definitions and notations. In Sec. 3.1, we give a concise description of digital topology and simple points. Next, we analyze different types of warping errors in Sec. 3.2. The proposed warping loss is introduced in Sec. 3.3. And finally, we explain the proposed new warping algorithm in Sec. 3.4.

### 3.1 Digital Topology and Simple Points

In this section, we briefly introduce simple point definition from the classic digital topology [30]. We focus on the 2D setting, whereas all definitions generalize to 3D. Details on 3D images will be provided in the Sec. A.2.

**Connectivities of pixels.** To discuss the topology of a 2D binary image, we first define the connectivity between pixels. See Fig. 3 for an illustration. A pixel $p$ has 8 pixels surrounding it. We can either consider the 4 pixels that share an edge with $p$ as $p$'s neighbors (called *4-adjacency*), or consider all 8 pixels as $p$'s neighbors (called *8-adjacency*). To ensure the Jordan closed curve theorem to hold, one has to use one adjacency for foreground (FG) pixels, and the other adjacency for the background (BG) pixels. In this paper, we use 4-adjacency for FG and 8-adjacency for BG. For 3D binary images, we use 6-adjacency for FG and 26-adjacency for BG. Denote by $N_4(p)$ the set of 4-adjacency neighbors of $p$, and $N_8(p)$ the set of 8-adjacency neighbors of $p$.

**Simple points.** For a binary image (2D/3D), a pixel/voxel is called a *simple point* if it could be flipped from foreground (FG) to background (BG), or from BG to FG, without changing the topology of the image [30]. The following definition can be used to determine whether a point is simple:

**Definition 1** (Simple Point Condition [30]) *Let $p$ be a point in a 2D binary image. Denote by $F$ the set of FG pixels. Assume 4-adjacency for FG and 8-adjacent for BG. $p$ is a simple point if and only if both of the following conditions hold: 1) $p$ is 4-adjacent to just one FG connected component in $N_8(p)$; and 2) $p$ is 8-adjacent to just one BG connected component in $N_8(p)$.*

See Fig. 3 for an illustration of simple and non-simple points in 2D case. It is easy to check if a pixel $p$ is simple or not by inspecting its $3 \times 3$ neighboring patch. The **Definition 1** can also generalize to 3D setting with 6- and 26-adjacencies for FG and BG, respectively.

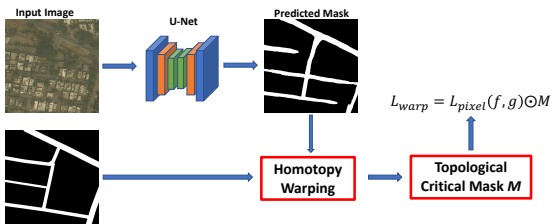

$$L_{warp} = L_{pixel}(f, g) \odot M$$

Figure 2: The illustration of the proposed *homotopy warping loss* $L_{warp}$. The homotopy warping algorithm tries to identify the topological critical pixels via the binary mask instead of noisy likelihood maps. These identified topological critical pixels/mask are used to define a new loss which is complementary to standard pixel-wise loss functions. The details of *Homotopy Warping* and *Topological Critical Mask M* can be found in Sec. 3.2 and Sec. 3.3, respectively.

## 3.2 Homotopic Warping Error

In this section, we introduce the homotopic warping of one mask towards another. We warp a mask through a sequence of flipping of simple points. Since we only flip simple points, by definition the warped mask will have the same topology.[1] The operation is called a *homotopic warping*. It has been proven that two binary images with the same topology can always be warped into each other by flipping a sequence of simple points [40].

Consider two input masks, the prediction mask and the ground truth mask. We can warp one of them (source mask)

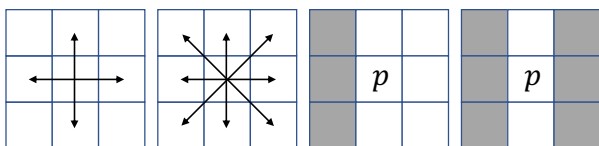

(a) 4-adjacent  (b) 8-adjacent (c) Simple Point(d) Non-simple

Figure 3: Illustration for 4, 8-adjacency, simple and non-simple points. **(a)**: 4-adjacency. **(b)**: 8-adjacency. **(c)**: a simple point $p$. White and grey pixels are FG and BG, respectively. Flipping the label of $p$ will not change the topology. **(d)**: a non-simple point $p$. Flipping $p$ will change the topology.

into another (target mask) in the best way possible, i.e., the warped mask has the minimal number of difference with the target mask (formally, the minimal Hamming distance). Once the warping is finished, the pixels at which the warped mask is different from the target mask, called *critical pixels*, are a sparse set of pixels indicative of the topological errors of the prediction mask.

We will warp in both directions: from the prediction mask to the ground truth mask, and the opposite. They identify different sets of critical pixels for a same topological error. In Fig. 4, we show a

---

[1]Note that it is essential to flip these simple points sequentially. The simple/non-simple status of a pixel may change if other adjacent pixels are flipped. Therefore, flipping a set of simple points *simultaneously* is not necessarily topology-preserving.

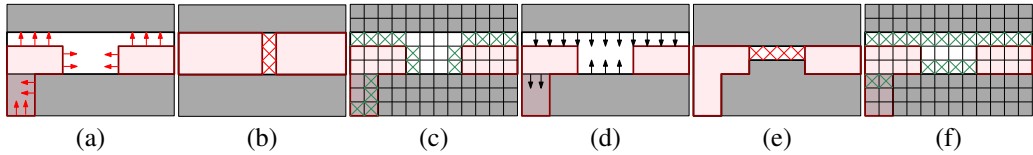

|  (a)  |  (b)  |  (c)  |  (d)  |  (e)  |  (f)  |

Figure 4: Illustration of homotopic warping between two masks, red and white. If red is the FG of the prediction, this is a false negative topological error; if red is the FG of the ground truth, this is a false positive error. **(a-c):** warping the red mask towards the white. **(a):** arrows show the warping direction. **(b):** the final mask after warping. Only a single-pixel wide gap remains in the middle of the warped red mask. The non-simple/critical pixels are highlighted with red crosses. They correspond to the topological error and will be penalized in the loss. **(c):** at the beginning of the warping, we highlight (with green crosses) simple points that can be flipped according to our algorithm. **(d-f):** warping the white mask towards the red mask. Only a single-pixel wide connection remains to ensure the warped white mask is connected. The non-simple/critical pixels are highlighted with the red crosses.

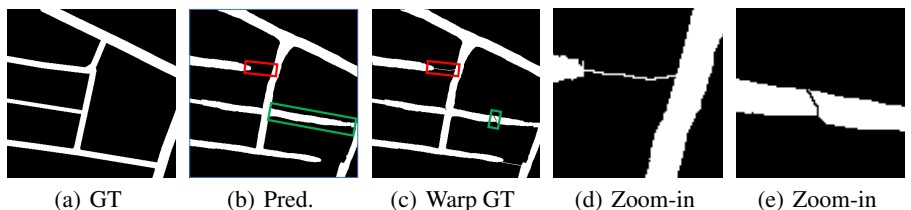

| (a) GT | (b) Pred. | (c) Warp GT | (d) Zoom-in | (e) Zoom-in |

Figure 5: Illustration of warping in a real world example (satellite image). **(a)** GT mask. **(b)** The prediction mask. The red box highlights a *false negative connection*, and the green box highlights a *false positive connection*. **(c)** Warped GT mask (using the prediction mask as the target). **(d)** Zoomed-in view of the red box in **(c)**. **(e)** Zoomed-in view of the green box in **(c)**.

synthetic example with red and white masks, as well warping in both directions. Warping the red mask towards the white mask ((a) and (b)) results in a single-pixel wide gap. The pixels in the gap (highlighted with red crosses) are critical pixels; flipping any of them will change the topology of the warped red mask. Warping the white mask towards the red mask ((d) and (e)) results in a single pixel wide link connecting the warped white mask. All pixels along the link (highlighted with red crosses) are critical; flipping any of them will change the topology of the warped white mask.

Here if the red mask is the prediction mask, then this corresponds to a false negative connection, i.e., a connection that is missed by the prediction. If the red mask is the ground truth mask, then this corresponds to a false positive connection. Note the warping ensures that *only topological errors are represented by the critical pixels*. In the synthetic example (Fig. 4), the large area of error in the bottom left corner of the image is completely ignored as it is not topologically relevant.

In Fig. 5, we show a real example from the satellite image dataset, focusing on the errors related to 1D topological structures (connection). In the figure we illustrate both a false negative connection error (highlighted with a red box) and false positive connection error (highlighted with a green box). If we warp the ground truth mask towards the prediction mask (c), we observe critical pixels forming a link for the false negative connection (d), and a gap for the false positive connection (e). Similarly, we can warp the prediction mask towards the ground truth, and get different sets of critical pixels for the same topological errors (illustrations will be provided in Sec. A.3).

Note that for 2D images with fine structures, errors regarding 1D topological structures are the most crucial. They affect the connectivity of the prediction results. For 3D images, errors on 1D or 2D topological structures are both important, corresponding to broken connections for tubular structures and holes in membranes. We will provide more comprehensive characterization of different types of topological structures and errors in Sec. A.4.

### 3.3 Homotopy Warping Loss

Next, we formalize the proposed *homotopy warping loss*, which is evaluated on the critical pixels due to homotopic warping. As illustrated in the previous section, the warping can be in both directions, from the prediction mask to the ground truth mask, and the opposite.

Formally, we denote by $f$ the predicted likelihood map of a segmentation network, and $f_B$ the corresponding binarized prediction mask (i.e., $f$ thresholded at 0.5). We denote by $g$ the ground truth mask. First, we warp $g$ towards $f_B$, so that the warped mask, $g^*$ has the minimal Hamming distance from the target $f_B$.

$$g^* = \arg\min_{g^w \lhd g} ||f_B - g^w||_H \tag{1}$$

where $\lhd$ is the homotopic warping operation. The pixels at which $g^*$ and $f_B$ disagree are the critical pixels and will be penalized in the loss. We record these critical pixels due to the warping of $g$ with a mask $M_g$, $M_g = f_B \oplus g^*$, in which $\oplus$ is the *Exclusive Or* operation.

We also warp the prediction mask $f_B$ towards $g$.

$$f_B^* = \arg\min_{f_B^w \lhd f_B} ||f_B^w - g||_H \tag{2}$$

We use the mask $M_f$ to record the remaining critical pixels after warping $f_B$, $M_f = g \oplus f_B^*$.

The union of the two critical pixel masks is the complete set of critical pixels corresponding to topological errors, $M = M_g \cup M_f$. $M$ contains all the locations directly related to topological structures. Note this is different from persistent-homology-based method [24], DMT based method [25] or skeleton based method [42], which extract topological locations/structures on the predicted continuous likelihood maps. Our warping loss directly locates the topological critical pixels/structures/locations based on the binary mask. The detected critical pixel set is sparse and less noisy.

$L_{pixel}$ denotes the pixel-wise loss function (e.g., cross-entropy), then $L_{warp}$ can be defined as:

$$L_{warp} = L_{pixel}(f, g) \odot M \tag{3}$$

where $\odot$ denotes Hadamard product. $L_{warp}$ penalizes the topological critical pixels, forcing the neural network to predict better at these locations, and thus are less prone to topological errors.

The final loss of our method, $L_{total}$, is given by:

$$L_{total} = L_{dice} + \lambda_{warp} L_{warp} \tag{4}$$

where $L_{dice}$ denotes the dice loss. And the loss weight $\lambda_{warp}$ is used to balance the two loss terms.

### 3.4 Distance-Ordered Homotopy Warping

Even though checking whether a pixel is simple or not is easy, finding the optimal warping as in Eq. (1) and (2) is challenging. The reason is that there are too many degrees of freedom. At each iteration during the warping, we have to choose a simple point to flip. It is not obvious which simple point will finally lead to a global optimum.

In this section, we provide an efficient heuristic algorithm to find a warping local optimum. We explain the algorithm for warping $g$ towards $f_B$. The algorithm generalizes to the opposite warping direction naturally. Recall the warping algorithm iteratively flips simple points. But there are too many choices at each iteration. It is hard to know which flipping choice will lead to the optimal solution. We need good heuristics for choosing a flippable pixel. Below we explain our main intuitions for designing our algorithm.

First, we restrict the warping so it only sweeps through the area where the two masks disagree. In other words, at each iteration, we restrict the candidate pixels for flipping to not only simple, but also pixels on which $g$ and $f_B$ disagree. In Fig. 4 (c) and (f), we highlight the candidate pixels for flipping at the beginning. Notice that not all simple points are selected as candidates. We only choose simple points within the difference set $\text{Diff}(f_B, g) = f_B \oplus g$.

Second, since we want to minimize the difference of the warped and target masks, we propose to flip pixels within the difference region $\text{Diff}(f_B, g)$. To implement this strategy efficiently, we order all pixel within $\text{Diff}(f_B, g)$ according to their distance from the FG/BG, and flip them according to this order. A pixel is skipped if it is not simple.

Our algorithm is based on the intuition that a far-away pixel will not become simple until nearby pixels are flipped first. To see this, we first formalize the definition of *distance transform* from the masks, $f_B$ and $g$, denoted by $D^{f_B}$ and $D^g$. For a BG pixel of $g$, $p$, its distance value $D^g(p)$ is the shortest distance from $p$ to any FG pixel of $g$, $D^g(p) = \min_{s \in FG_g} \text{dist}(p, s)$. Similarly, for a FG pixel of $g$, $q$, $D^g(q) = \min_{s \in BG_g} \text{dist}(q, s)$. The definition generalizes to $D^{f_B}$.

We observe that a pixel cannot be simple unless it has distance 1 from the FG/BG of a warping mask. The proof is straightforward. Formally,

**Lemma 1.** *Given a 2D binary mask $m$, a pixel $p$ cannot be simple for $m$ if its distance function $D^m(p) > 1$.*

**Proof.** Assume the foreground has a pixel value of 1 and $p$ is a background pixel with a index of $(i, j)$. Consider the $m$-adjacent ($m$=4) for $p$. Since $D^m(p) > 1$, then we have $m(i-1, j) = m(i+1, j) = m(i, j-1) = m(i, j+1) = 0$. In this case, $p$ is not 4-adjacent to any FG connected component, violating the **1)** of **Definition 1**. Consequently, pixel $(i, j)$ is not a simple point. This also holds for foreground pixels. This lemma naturally generalizes to 3D case. □

Lemma 1 implies that only after flipping pixels with distance 1, the other misclassified locations should be considered. This observation gives us the intuition of our algorithm. To warp $g$ towards $f_B$, our algorithm is as follows: (1) compute the difference set $\text{Diff}(f_B, g)$ as the candidate set of pixels; (2) sort candidate pixels in a non-decreasing order of the distance transform $D^g$; (3) enumerate through all candidate pixels according to the order. For each iteration, check if it is simple. If yes, flip the pixel's label. It is possible that this algorithm can miss some pixels. They are not simple when the algorithm checks, but they might become simple as the algorithm continues (since their neighboring pixels get flipped in previous iterations).

One remedy is to recalculate the distance transform after one round of warping, and go through remaining pixels once more. But in practice we found this is not necessary as this scenario is very rare. The effectiveness of the proposed warping strategy is illustrated in Sec. A.5.

## 4 Experiments

We conduct extensive experiments to demonstrate the effectiveness of the proposed method. Sec. A.6 introduces the datasets used in this paper, including both 2D and 3D datasets. The benchmark methods are described in Sec. A.7. We mainly focus on topology-aware segmentation methods. Sec. A.8 describes the evaluation metrics used to assess the quality of the segmentation. To demonstrate the ability to achieve better structural/topological performances, besides standard segmentation metric, such as DICE score, we also use several topology-aware metrics to evaluate all the methods. Several ablation studies are then conducted to further demonstrate the efficiency and effectiveness of the technical contributions (Sec. 4.1).

**Datasets.** We conduct extensive experiments to validate the efficacy of our method. Specifically, we use four natural and biomedical 2D datasets (**RoadTracer** [4], **DeepGlobe** [12], **Mass.** [34], **DRIVE** [44]) and one more 3D biomedical dataset (**CREMI** [2]) to

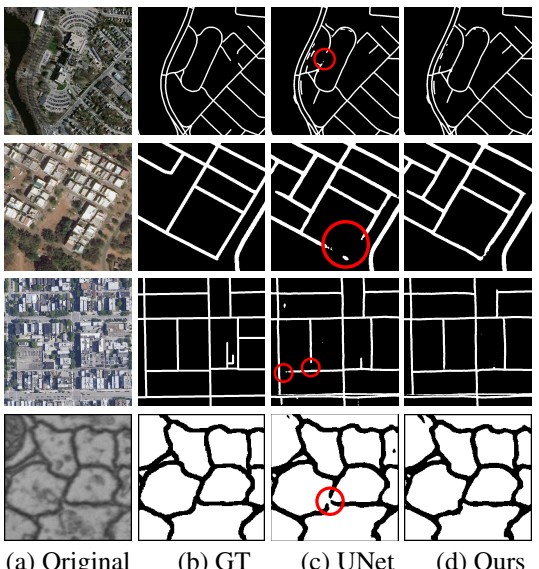

(a) Original (b) GT (c) UNet (d) Ours

Figure 6: Qualitative results compared with the standard UNet. The proposed warping loss can help to correct the topological errors (highlighted by red circles). The sampled patches are from four different datasets.

validate the efficacy of the propose method. More details about the datasets and the split of the training and validation subsets are included in Sec. A.6.

**Baselines.** We compare the results of our method with several state-of-the-art methods. The standard/simple UNet (2D/3D) **UNet** [39, 10] is used as a strong baseline and the backbone for other methods. The other baselines used in this paper include **RoadTracer** [4], **VecRoad** [48], **iCurb** [52],

[2]https://cremi.org/

**DIVE** [17], **VGG-UNet** [35], **TopoLoss** [24], **clDice** [42] and **DMT** [25]. More descriptions and implementation details of these baselines are included in Sec. A.7.

**Evaluation Metrics.** We use both pixel-wise (**DICE**) and topology-aware metrics (**ARI**, **Warping Error** and **Betti number error**) to evaluate the performance of the proposed method. More details about the evaluation metrics are provided in Sec. A.8.

**Implementation Details.** For 2D images, we use $(m, n) = (4, 8)$ to check if a pixel is simple or not; while $(m, n) = (8, 26)$ for 3D images. We choose cross-entropy loss as $L_{warp}/L_{pixel}$ for all the experiments, except the ablation studies in Tab. 3.

For 2D datasets, the batch size is set as 16, and the initial learning rate is 0.01. We randomly crop patches with the size of $512 \times 512$ and then feed them into the 2D UNet. For 3D case, the batch size is also 16, while the input size is $128 \times 128 \times 16$. We perform the data normalization for the single patch based on its mean and standard deviation.

We use PyTorch framework (Version: 1.7.1) to implement the proposed method. A simple/standard UNet (2D or 3D) is a used as baseline and the backbone. For the proposed method, as well as the other loss function based baselines, to make a fair comparison, we use the same UNet as backbone. And the training strategy is to train the UNet with dice loss first until converge, and then add the proposed losses to fine-tune the models obtained from the initial step.

All the experiments are performed on a Tesla V100-SXM2 GPU (32G Memory), and an Intel(R) Xeon(R) Gold 6140 CPU@2.30 GHz.

**Qualitative and Quantitative Results.** In Fig. 6, we show qualitative results from different datasets. Compared with baseline UNet, our method recovers better structures, such as connections, which are highlighted by red circles. Our final loss is a weighted combination of the dice loss and warping-loss term $L_{warp}$. When $\lambda_{warp} = 0$, the proposed method degrades to a standard UNet. The recovered better structures (UNet and *Warping* columns in Fig. 6) demonstrates that our warping-loss helps the deep neural networks to achieve better topological segmentations. More qualitative results are provided in Sec. A.9.

Table 1: Quantitative results of different methods.

| Method | DICE↑ | ARI↑ | Warping↓ | Betti↓ |
|---|---|---|---|---|
| RoadTracer | | | | |
| UNet [39] | 0.587 | 0.544 | $10.412 \times 10^{-3}$ | 1.591 |
| RoadTracer [4] | 0.547 | 0.521 | $13.224 \times 10^{-3}$ | 2.218 |
| VecRoad [48] | 0.552 | 0.533 | $12.819 \times 10^{-3}$ | 2.095 |
| iCurb [52] | 0.571 | 0.535 | $11.683 \times 10^{-3}$ | 1.873 |
| VGG-UNet [35] | 0.576 | 0.536 | $11.231 \times 10^{-3}$ | 1.607 |
| TopoNet [24] | 0.584 | 0.556 | $10.008 \times 10^{-3}$ | 1.378 |
| clDice [42] | 0.591 | 0.550 | $9.192 \times 10^{-3}$ | 1.309 |
| DMT [25] | 0.593 | 0.561 | $9.452 \times 10^{-3}$ | 1.419 |
| *Warping* | **0.603** | **0.572** | **8.853** $\times 10^{-3}$ | **1.251** |
| DeepGlobe | | | | |
| UNet [39] | 0.764 | 0.758 | $3.212 \times 10^{-3}$ | 0.827 |
| VGG-UNet [35] | 0.742 | 0.748 | $3.371 \times 10^{-3}$ | 0.867 |
| TopoNet [24] | 0.765 | 0.763 | $2.908 \times 10^{-3}$ | 0.695 |
| clDice [42] | 0.771 | 0.767 | $2.874 \times 10^{-3}$ | 0.711 |
| DMT [25] | 0.769 | 0.772 | $2.751 \times 10^{-3}$ | 0.609 |
| *Warping* | **0.780** | **0.784** | **2.683** $\times 10^{-3}$ | **0.569** |
| Mass. | | | | |
| UNet [39] | 0.661 | 0.819 | $3.093 \times 10^{-3}$ | 3.439 |
| VGG-UNet [35] | 0.667 | 0.846 | $3.185 \times 10^{-3}$ | 2.781 |
| TopoNet [24] | 0.690 | 0.867 | $2.871 \times 10^{-3}$ | 1.275 |
| clDice [42] | 0.682 | 0.862 | $2.552 \times 10^{-3}$ | 1.431 |
| DMT [25] | 0.706 | **0.881** | $2.631 \times 10^{-3}$ | 0.995 |
| *Warping* | **0.715** | 0.864 | **2.440** $\times 10^{-3}$ | **0.974** |
| DRIVE | | | | |
| UNet [39] | 0.749 | 0.834 | $4.781 \times 10^{-3}$ | 3.643 |
| DIVE [17] | 0.754 | 0.841 | $4.913 \times 10^{-3}$ | 3.276 |
| VGG-UNet [35] | 0.721 | 0.887 | $4.362 \times 10^{-3}$ | 2.784 |
| TopoNet [24] | 0.761 | 0.902 | $3.895 \times 10^{-3}$ | 1.076 |
| clDice [42] | 0.753 | 0.896 | $4.012 \times 10^{-3}$ | 1.218 |
| DMT [25] | 0.773 | 0.908 | $3.561 \times 10^{-3}$ | 0.873 |
| *Warping* | **0.781** | **0.911** | **3.419** $\times 10^{-3}$ | **0.812** |
| CREMI | | | | |
| 3D UNet [10] | 0.961 | 0.832 | $11.173 \times 10^{-3}$ | 2.313 |
| DIVE [17] | 0.964 | 0.851 | $11.219 \times 10^{-3}$ | 2.674 |
| TopoNet [24] | 0.967 | 0.872 | $10.454 \times 10^{-3}$ | 1.076 |
| clDice [42] | 0.965 | 0.845 | $10.576 \times 10^{-3}$ | 0.756 |
| DMT [25] | **0.973** | 0.901 | $10.318 \times 10^{-3}$ | 0.726 |
| *Warping* | 0.967 | **0.907** | **9.854** $\times 10^{-3}$ | **0.711** |

Tab. 1 shows quantitative results for three 2D image datasets, RoadTracer, DeepGlobe and The Massachusetts dataset and one 3D image dataset, CREMI. The best performances are highlighted with bold. The proposed warping-loss usually achieves the best performances in both DICE score and topological accuracy (ARI, Warping Error and Betti Error) over other topology-aware segmentation baselines. Note that we only report the mean performances here for space limitation. We also include standard deviations and use t-test to determine the statistical significance in Sec. A.10.

### 4.1 Ablation studies

To further explore the technical contributions of the proposed method and provide a rough guideline of how to choose the hyperparameters, we conduct several ablation studies. Note that all the ablation studies are conducted on the RoadTracer dataset.

**The impact of the loss weights.** As seen in Eq. 4, our final loss function is a combination of dice loss and the proposed warping loss $L_{warp}$. The balanced term $\lambda_{warp}$ controls the influence of the warping loss term, and it's a dataset dependent hyper-parameter. The quantitative results for different choices of $\lambda_{warp}$ are illustrated in Tab. 2. For the RoadTracer dataset, the optimal value is $1 \times 10^{-4}$. From Tab. 2, we can find that different choices of $\lambda_{warp}$ do affect the performances. The reason is that, if $\lambda_{warp}$ is too small, the effect of the warping loss term is negligible. However, if $\lambda_{warp}$ is too large, the warping loss term will compete with the $L_{dice}$ and decrease the performance of the other easy-classified pixels. Note that within a reasonable range of $\lambda_{warp}$, all the choices contribute to better performances compared to baseline (row '0', standard UNet), demonstrating the effectiveness of the proposed loss term. The best choices of $\lambda_{warp}$ for each specific dataset are included in Sec. A.11.

Table 2: Ablation study for loss weight $\lambda_{warp}$.

| $\lambda_{warp}$ | DICE↑ | ARI↑ | Warping↓ | Betti↓ |
|---|---|---|---|---|
| 0 | 0.587 | 0.544 | 10.412 $\times10^{-3}$ | 1.591 |
| $2 \times 10^{-5}$ | **0.603** | 0.561 | 9.012 $\times10^{-3}$ | 1.307 |
| $5 \times 10^{-5}$ | 0.601 | 0.548 | 9.356 $\times10^{-3}$ | 1.412 |
| $1 \times 10^{-4}$ | **0.603** | **0.572** | **8.853** $\times10^{-3}$ | **1.251** |
| $2 \times 10^{-4}$ | 0.602 | 0.565 | 9.131 $\times10^{-3}$ | 1.354 |

**The choice of loss functions.** The proposed warping loss is defined on the identified topological critical pixels. Consequently, any pixel-wise loss functions can be used to define the warping loss $L_{warp}/L_{pixel}$. In this section, we investigate how the choices of loss functions affect the performances. The quantitative results are show in Tab. 3. Compared with mean-square-error loss (MSE) or Dice loss, the cross-entropy loss (CE) achieves best performances in terms of topological metrics. On the other hand, all these three choices perform better than baseline method (row 'w/o', standard UNet), which further demonstrates the contribution of the proposed loss term.

Table 3: Ablation study for the choices of loss.

| $L_{pixel}$ | DICE↑ | ARI↑ | Warping↓ | Betti↓ |
|---|---|---|---|---|
| w/o | 0.587 | 0.554 | 10.412 $\times10^{-3}$ | 1.591 |
| MSE | 0.598 | 0.556 | 9.853 $\times10^{-3}$ | 1.429 |
| Dice loss | **0.606** | 0.563 | 9.471 $\times10^{-3}$ | 1.368 |
| CE | 0.603 | **0.572** | **8.853** $\times10^{-3}$ | **1.251** |

**Comparison of different critical pixel selection strategies.** We also conduct additional experiments to demonstrate the effectiveness of the proposed critical pixel selection strategy. The first variation is flipping the simple pixels but removing the heuristic that uses the distance transform, and then use the remaining non-simple pixels as our critical pixel set, which is denoted as 'w/o' in Tab. 4.

Table 4: Comparison of different critical pixel selection strategies.

| Kernel Size | DICE↑ | ARI↑ | Warping ↓ | Betti↓ |
|---|---|---|---|---|
| U-Net | 0.587 | 0.544 | 10.412 $\times10^{-3}$ | 1.591 |
| w/o DT | 0.586 | 0.547 | 10.256$\times10^{-3}$ | 1.473 |
| Warping (GT → Pred) | 0.594 | 0.567 | 9.171$\times10^{-3}$ | 1.290 |
| Warping (Pred → GT) | 0.598 | 0.562 | 9.124 $\times10^{-3}$ | 1.315 |
| *Warping* | **0.603** | **0.572** | **8.853**$\times10^{-3}$ | **1.251** |

The other two variations (warping only in one direction: ground truth to prediction or prediction to ground truth) achieve reasonably better while still slightly inferior results to the proposed version. The reason might be that the proposed critical point selection strategy contains more complete topologically challenging pixels. Both ablation studies demonstrate the effectiveness of the proposed critical point selection strategy.

**Comparison with morphology post-processing.** To verify the necessity of the proposed method, we also compare the proposed method with traditional morphology post-processing, i.e., dilation, erosion. Dilation and erosion are global operations. Though closing operation (dilates image and then erodes dilated image) could bridge some specific gaps/broken connections, it will damage the global structures. In practice, the gaps/broken will usually be more than a few pixels. If the kernel size is too small, the closing operation (dilate then erode) will hardly affect the final performance; while too big kernel sizes will join the separated regions. Tab. 5 lists post-processing results on baseline U-Net.

**The efficiency of the proposed loss.** In this section, we'd like to investigate the efficiency of the proposed method. Our warping algorithm contains two parts, the distance transform and the sorting of distance matrix. The complexity for distance transform is $O(n)$ for a 2D image where $n$ is the size of the 2D image, and $n = H \times W$. $H, W$ are the height and width of the 2D image, respectively. And the complexity for sorting is $mlog(m)$, where $m$ is the number of

Table 5: Comparison against post-processing.

| Kernel Size | DICE↑ | ARI↑ | Warping ↓ | Betti↓ |
|---|---|---|---|---|
| U-Net | 0.587 | 0.544 | $10.412 \times 10^{-3}$ | 1.591 |
| Closing (5) | 0.588 | 0.542 | $10.414 \times 10^{-3}$ | 1.590 |
| Closing (10) | 0.587 | 0.546 | $10.399 \times 10^{-3}$ | 1.583 |
| Closing (15) | 0.586 | 0.541 | $10.428 \times 10^{-3}$ | 1.598 |
| *Warping* | **0.603** | **0.572** | $\mathbf{8.853} \times 10^{-3}$ | **1.251** |

misclassified pixels. Usually $m \ll n$, so the overall complexity for the warping algorithm is $O(n)$. As a comparison, [24] needs $O(n^3)$ complexity to compute the persistence diagram. And the computational complexity for [25], [42] are $O(nlog(n))$ and $O(n)$, respectively.

The comparison in terms of complexity and training time are illustrated in Tab. 6. Note that for the proposed method and all the other baselines, we first train a simple/standard UNet, and then add the additional loss terms to fine-tune the models obtained from the initial step. Here, the training time is only for the fine-tune step. The proposed method takes slightly longer training time than clDice, while achieves the best performance over the others. As all these methods use the same backbone, the inference times are the same.

Table 6: Comparison of efficiency.

| Method | Complexity | Training time |
|---|---|---|
| TopoNet [24] | $O(n^3)$ | $\approx 12h$ |
| clDice [42] | $O(n)$ | $\approx \mathbf{3h}$ |
| DMT [25] | $O(nlogn)$ | $\approx 7h$ |
| *Warping* | $O(n)$ | $\approx 4h$ |

## 5 Conclusion

In this paper, we propose a novel homotopy warping loss to learn to segment with better structural/topological accuracy. Under the homotopy warping strategy, we can identify the topological critical pixels/locations, and the new loss is defined on these identified pixels/locations. Furthermore, we propose a novel strategy called Distance-Ordered Homotopy Warping to efficiently identify the topological error locations based on distance transform. Extensive experiments on multiple datasets and ablation studies have been conducted to demonstrate the efficacy of the proposed method.

**Limitations.** We mainly focus on the binary segmentation of curvilinear structures in this work. In terms of multiclass curvilinear structure segmentation, we note that it's always doable to convert the multiclass segmentation task to a number of binary class segmentation tasks, which will be left for future work.

**Acknowledgement.** The research of Xiaoling Hu is partially supported by NSF IIS-1909038, and we would like to thank the anonymous reviews for constructive comments.

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
