# Structure-Aware Image Segmentation with Homotopy Warping
## — Supplementary Material—

**Xiaoling Hu**

Department of Computer Science
Stony Brook University
xiaolhu@cs.stonybrook.edu

## A  Appendix

Appendix A.1 shows the comparison of critical points identified by TopoNet [7] and the proposed homotopy warping.

Appendix A.2 illustrates the *simple points* in 3D case.

Appendix A.3 shows the results of warping the prediction binary mask towards GT mask.

Appendix A.4 illustrates the topological errors in 3D case.

Appendix A.5 illustrates the effectiveness of the proposed warping strategy.

Appendix A.6 provides the details of the datasets used in this paper.

Appendix A.7 describes the details of the baselines used in this paper.

Appendix A.8 illustrates the details of the metrics used in this paper.

Appendix A.9 shows more qualitative results from different datasets.

Appendix A.10 provides stddev besides mean, and we use t-test to determine the statistical significance.

Appendix A.11 illustrates the loss weight (Cross-Entropy loss) for each dataset.

Appendix A.12 shows a few failure cases.

### A.1   Comparison of critical points between TopoNet [7] and Homotopy Warping

Compared to the proposed method (Fig. 7(f)), the critical points identified from [7] are very noisy and often are not relevant to the topological errors.

### A.2   Simple points in 3D

For a 3D binary image, a voxel is called a *simple point* if it could be flipped from foreground (FG) to background (BG), or from BG to FG, without changing the topology of the image [10]. The following definition is a natural extension of the **Definition 1** in the main paper. It can be used to determine whether a voxel is simple:

**Definition 2** (Simple Point Condition [10]) *Let $p$ be a point in a 3D binary image. Denote by $F$ the set of FG pixels. Assume 6-adjacency for FG and 26-adjacent for BG.*

- *$p$ is 6-adjacent to just one FG connected component in $N_{26}(p)$; and*

36th Conference on Neural Information Processing Systems (NeurIPS 2022).

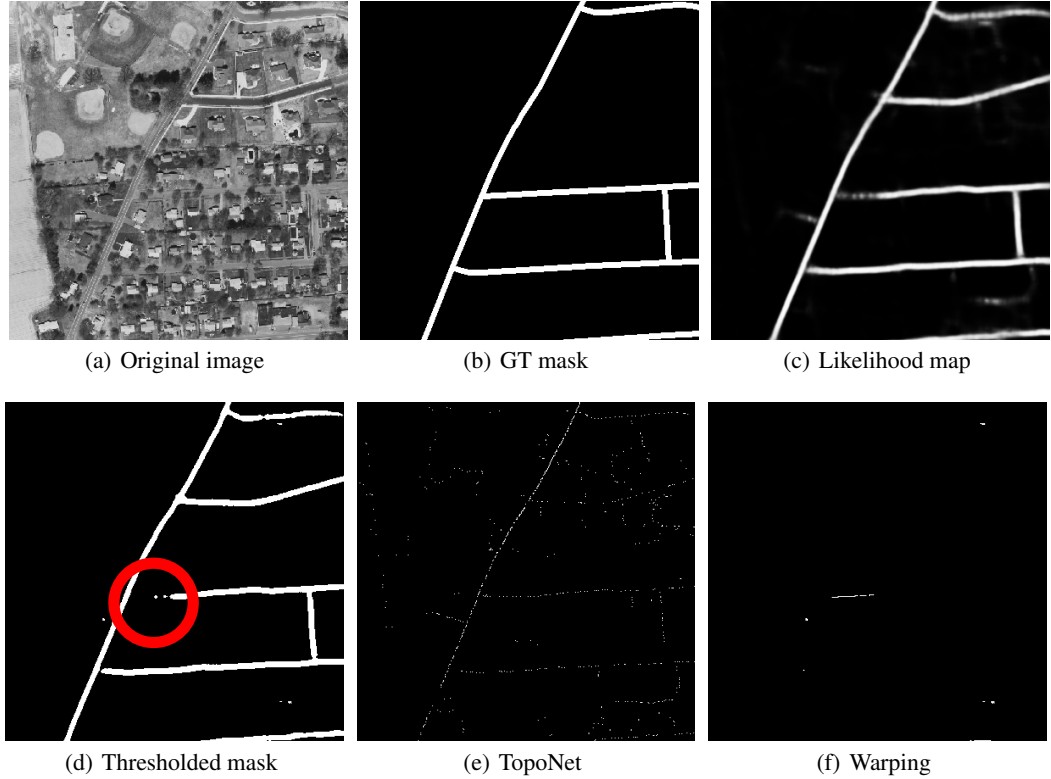

Figure 7: Illustration of the critical points identified by different methods. **(a)**: Original image. **(b)**: GT mask. **(c)**: Predicted likelihood map. **(d)**: Segmentation (Thresholded mask from likelihood map). **(e)**: Critical points identified by [7]. **(f)**: Critical points identified by our homotopy warping. Please zoom-in for better viewing.

- $p$ is 26-adjacent to just one BG connected component in $N_{26}(p)$.

### A.3 Illustration of warping prediction mask to GT

In Fig. 5 of the main paper, we provide the illustration of warping the GT mask towards prediction mask. Similarly, we can warp the prediction mask towards the GT mask, and get different sets of critical pixels for the same topological errors, which are shown in Fig. 9.

### A.4 Topological errors in 3D

In the main text, we provide the analysis of 1D topological structures (connections). Here we provide the illustration of 2D topological structures (holes/voids) for 3D case, which is illustrated in Fig. 10.

Note that for 3D vessel data, it's also the 1D topological structures (connections) that matter. The process of identifying the topological critical pixels is the same as Fig. 4 and Fig. 5 in the main text. And the only difference is to use 6-adjacency for FG and 26-adjacency for BG.

### A.5 Effectiveness of the proposed warping strategy

By using Distance-Ordered Homotopy Warping, we are able to only consider all the inconsistent pixels once and flip them if they are simple. Otherwise, we need to iteratively warp all the inconsistent pixels (one non-simple pixel might become simple if its neighbors are flipped in the previous iteration). It takes 1.452s to warp a 512×512 image without the warping strategy, while only 0.317s with the strategy thereby allowing the network to converge much faster.

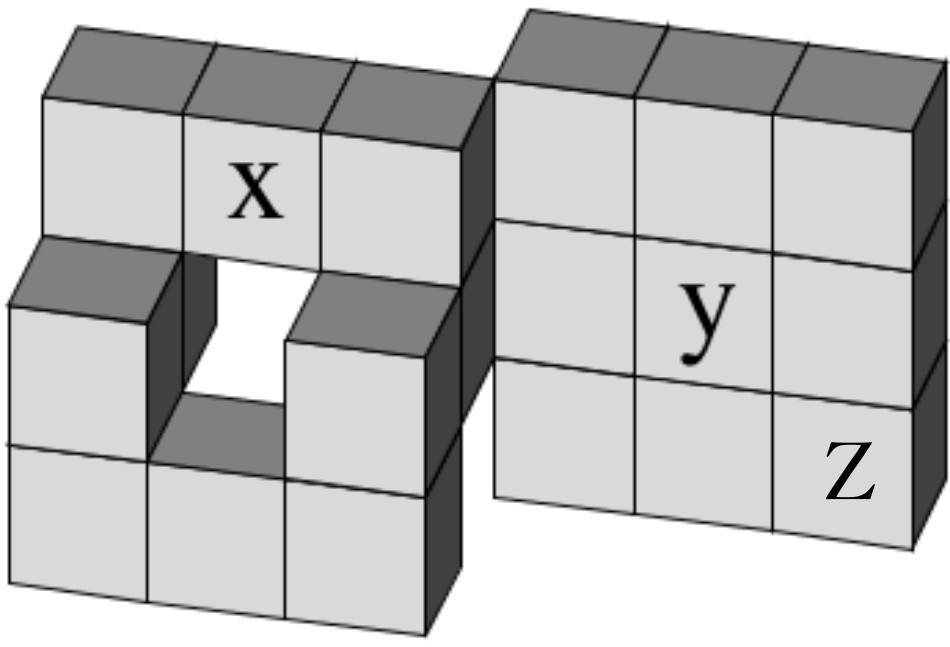

Figure 8: Illustration of *simple points* in 3D case. This figure is a 3D grid binary mask. Image credit to [4]. In this case, voxels $X$ and $Y$ are non-simple points/voxels, while $Z$ is a simple point/voxel.

## A.6 Datasets

The details of the datasets used in the paper are listed as follows:

1. *RoadTracer*: Roadtracer contains 300 high resolution satellite images, covering urban areas of forty cities from six different countries [1]. Similar to setting in [1], twenty five cities (180 images) are used as training set and the rest fifteen cities (120 images) are used as the validation set.

2. *DeepGlobe*: DeepGlobe contains aerial images of rural areas in Thailand, Indonesia and India [5]. Similar to setting in [2], we use 4696 images as training set and the rest 1530 images as validation set.

3. *Massachusetts*: The Massachusetts dataset [11] contains images from both urban and rural areas. Similar to the setting in [7], we conduct a three cross validation.

4. *DRIVE*: DRIVE [16] is a retinal vessel segmentation dataset with 20 images (resolution 584x565).

5. *CREMI*: The CREMI dataset is a 3D neuron dataset [1], whose resolution of $4 \times 4 \times 40$ nm. We also conduct a three cross validation.

## A.7 Baselines

The baseline methods used in this paper are listed as follows:

1. *RoadTracer* [1]: RoadTracer is an iterative graph construction based method where node locations are selected by a CNN.

2. *VecRoad* [17]: VecRoad is a point-based iterative graph exploration scheme with segmentation-cues guidance and flexible steps.

3. *iCurb* [18]: iCurb is an imitation learning-based solution for off-line road-curb detection method.

---

[1]https://cremi.org/

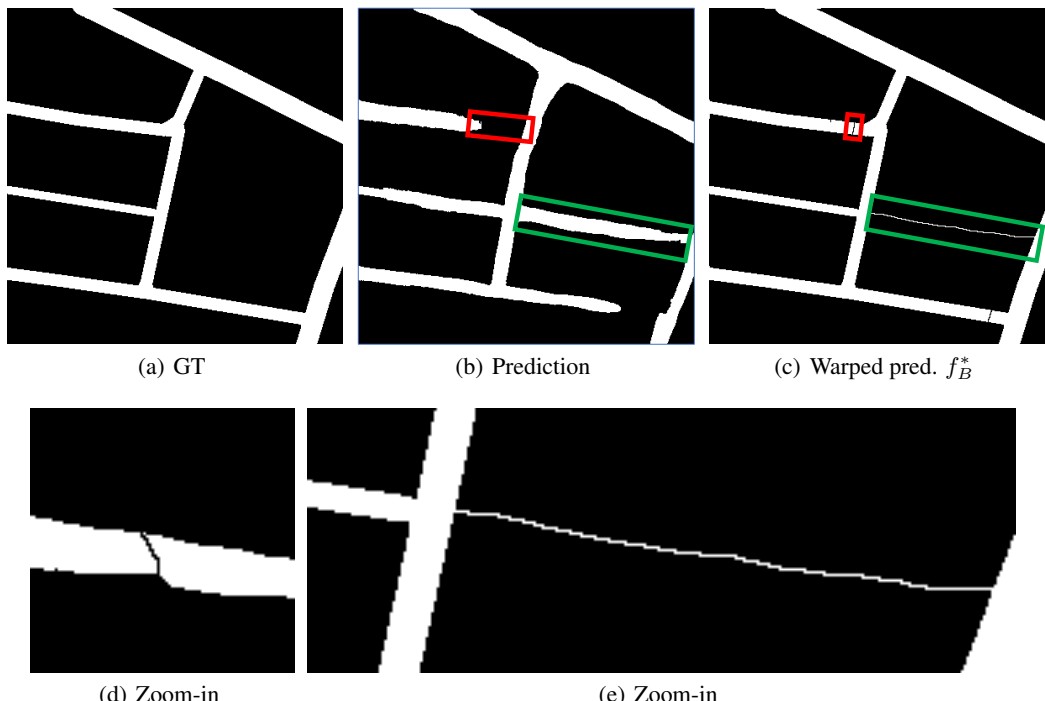

(a) GT       (b) Prediction       (c) Warped pred. $f_B^*$

(d) Zoom-in            (e) Zoom-in

Figure 9: Illustration of warping in a real world example (satellite image). **(a)** GT mask. **(b)** The prediction mask. The red box highlights a *false negative connection*, and the green box highlights a *false positive connection*. **(c)** Warped prediction mask (using the GT mask as the target). **(d)** Zoomed-in view of the red box in **(c)**. **(e)** Zoomed-in view of the green box in **(c)**.

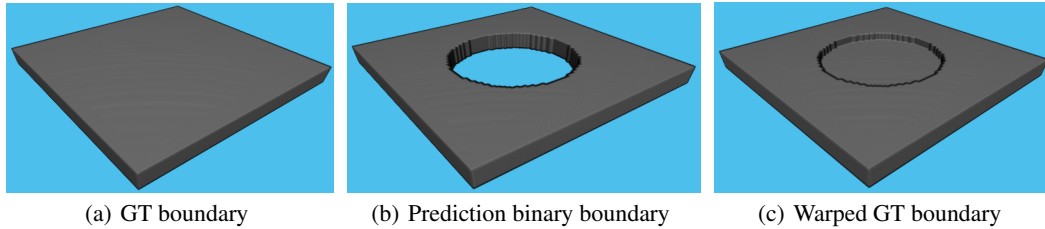

(a) GT boundary      (b) Prediction binary boundary      (c) Warped GT boundary

Figure 10: Illustration of 2D topological structures (holes/voids) for 3D case. **(a)**: GT boundary. **(b)**: Prediction binary boundary. **(c)**: Warped GT boundary. If we warp the GT boundary (Fig.**(a)**) towards prediction binary boundary (Fig.**(b)**), there will be a plane with a thickness of 1 in the middle of the hole/void to keep the original structure, which is illustrated in Fig. **(c)**.

4. *UNet* [14, 3]: The standard UNet trained with dice loss. Though lots of other segmentation methods/backbones have been proposed for image segmentation, UNet is still one of the most powerful methods for image segmentation with fine-structures.

5. *DIVE* [6]: DIVE is a popular EM neuron segmentation method.

6. *VGG-UNet* [12]: VGG-UNet uses the response of selected filters from a pretrained CNN to construct a new loss function. This is one of the earliest works trying to deal with correct delineation.

7. *TopoNet* [7]: TopoNet is a recent work which tries to learn to segment with correct topology based on a novel persistent homology based loss function.

8. *clDice* [15]: Another topology aware method for tubular structure segmentation. The basic idea is to use thinning techniques to extract the skeletons (centerlines) of the likelihood

maps and ground truth mask. A new cldice loss is proposed based on the extracted skeletons besides traditional pixel-wise loss.

9. *DMT* [8]: DMT is a topology-aware deep image segmentation method via discrete morse theory. Instead of identifying topological critical pixels/locations, the DMT loss tries to identify the whole morse structures, and the new loss is defined on the identified morse structures.

Note that *RoadTracer*, *VecRoad* and *iCurb* are graph-based methods for road tracing. Graph-based approaches learn to explicitly detect keypoints and connect them. Since the graph is built iteratively, some detection errors in the early stage can propagate and lead to even more errors. Segmentation-based methods avoid this issue as they make predictions in a global manner. The challenge with segmentation methods in road network reconstruction is they may fail in thin structures especially when the signal is weak. This is exactly what we are addressing in this paper – using critical pixels to improve segmentation-based methods.

*VGG-UNet*, *TopoNet*, *clDice* and *DMT* are topology-aware segmentation methods.

## A.8 Evaluation metrics

The details of the metrics used in this paper are listed as follows:

1. *DICE*: DICE score (also known as DICE coefficient, DICE similarity index) is one of the most popular evaluation metrics for image segmentation, which measures the overlapping between the predicted and ground truth masks.

2. *Adapted Rand Index (ARI)*: ARI is the maximal F-score of the foreground-restricted Rand index [13], a measure of similarity between two clusters. The intuition is that the boundaries partition the whole binary mask into several separate regions, and the predicted and ground truth binary masks can be regarded as two different partitions. ARI is used to measure the similarity between these two different partitions.

3. *Warping Error* [9]: Warping Error is metric that measures topological disagreements instead of simple pixel disagreements. After warping all the simple points of ground truth to the predicted mask, the disagreements left are topological errors. The warping error is defined as the percentage of these topological errors over the image size.

4. *Betti Error*: Betti Error directly calculates the topology difference between the predicted segmentation and the ground truth. We randomly sample patches over the predicted segmentation and compute the average absolute error between their Betti numbers and the corresponding ground truth patches.

## A.9 More qualitative results

We provide more qualitative results in Fig. 11. Compared with baseline UNet, our method recovers better structures, such as connections, which are highlighted by red circles. The recovered better structures (UNet and Warping columns in Fig. 11) demonstrates that our warping-loss helps the deep neural networks to achieve better topological segmentations.

## A.10 More quantitative results

We provide stddev besides mean, and use t-test (95% confidence interval) to determine the statistical significance (highlighted with bold in Tab. 7) for RoadTracer dataset. The quantitative results show that the proposed method performs significantly better than baselines.

## A.11 Loss weight parameter for each dataset

As illustrated in Sec. 4.1, we finally choose cross-entropy loss (CE) as $L_{pixel}/L_{warp}$ for all the datasets. In Tab. 8, we list the weights achieving the best performances on each dataset.

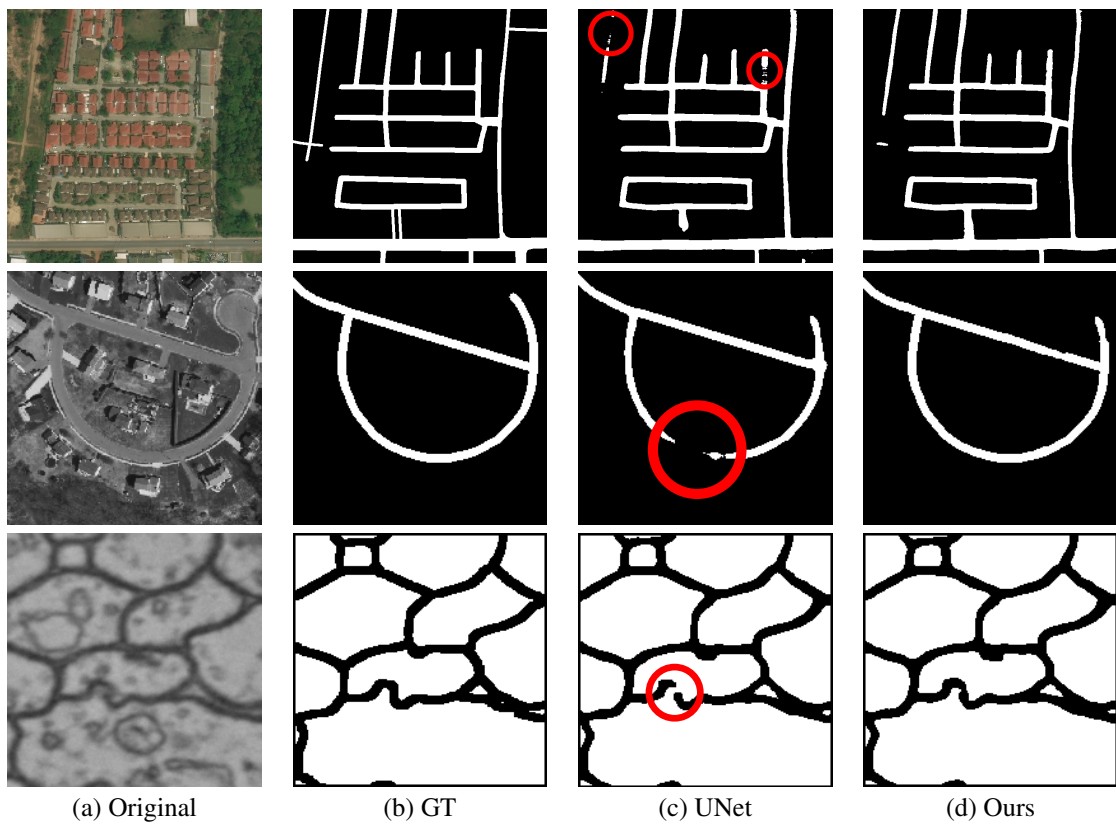

|  (a) Original | (b) GT | (c) UNet | (d) Ours |

Figure 11: More qualitative results compared with the standard UNet. The proposed warping loss can help to correct the topological errors (highlighted by red circles).

Table 7: Mean and stddev for different methods on RoadTracer.

| Method | DICE↑ | ARI↑ | Warping ($\times 10^{-3}$)↓ | Betti↓ |
|---|---|---|---|---|
| UNet [14] | $0.587 \pm 0.011$ | $0.544 \pm 0.006$ | $10.412 \pm 0.212$ | $1.591 \pm 0.098$ |
| RoadTracer | $0.547 \pm 0.012$ | $0.521 \pm 0.007$ | $13.224 \pm 0.429$ | $2.218 \pm 0.217$ |
| VecRoad | $0.552 \pm 0.009$ | $0.533 \pm 0.002$ | $12.819 \pm 0.173$ | $2.095 \pm 0.191$ |
| iCurb | $0.571 \pm 0.010$ | $0.535 \pm 0.008$ | $11.683 \pm 0.355$ | $1.873 \pm 0.104$ |
| VGG-UNet [12] | $0.576 \pm 0.004$ | $0.536 \pm 0.013$ | $11.231 \pm 0.183$ | $1.607 \pm 0.176$ |
| TopoNet [7] | $0.584 \pm 0.008$ | $0.556 \pm 0.010$ | $10.008 \pm 0.324$ | $1.378 \pm 0.075$ |
| clDice [15] | $0.591 \pm 0.005$ | $0.550 \pm 0.007$ | $9.192 \pm 0.209$ | $1.309 \pm 0.103$ |
| DMT [8] | $0.593 \pm 0.004$ | $0.561 \pm 0.002$ | $9.452 \pm 0.301$ | $1.419 \pm 0.092$ |
| *Warping* | $\mathbf{0.603 \pm 0.003}$ | $\mathbf{0.572 \pm 0.007}$ | $\mathbf{8.853 \pm 0.267}$ | $\mathbf{1.251 \pm 0.062}$ |

## A.12   Failure cases

In this section, we add a few failure cases from different datasets. Note that inferring topology given an image is a very difficult task, especially near challenging spots, e.g., blurred membrane locations or weak vessel connections. Current methods can help to improve the topology-wise accuracy, while they are far from perfect.

Table 8: Choice of loss weights.

| Dataset | RoadTracer | DeepGlobe | Mass | DRIVE | CREMI |
|---|---|---|---|---|---|
| $\lambda_{warp}$ | $1 \times 10^{-4}$ | $1 \times 10^{-4}$ | $1 \times 10^{-4}$ | $1 \times 10^{-4}$ | $2 \times 10^{-5}$ |

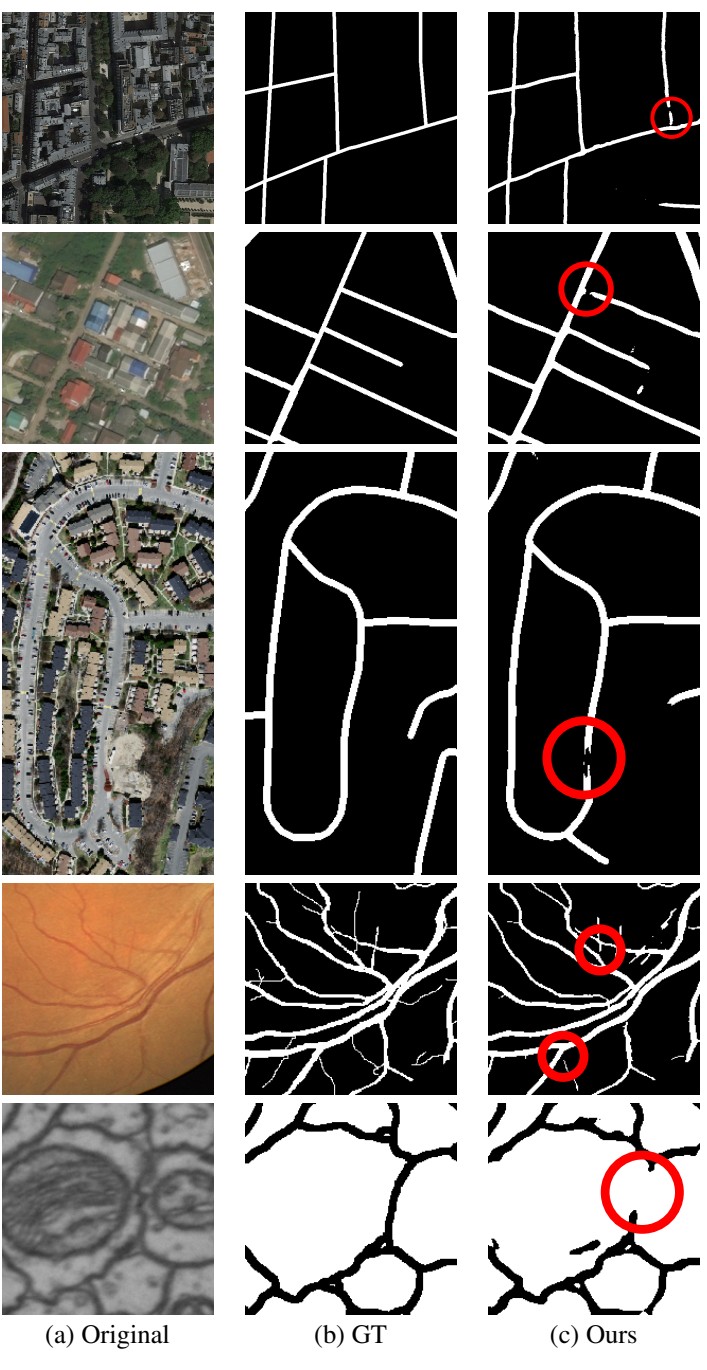

(a) Original      (b) GT      (c) Ours

Figure 12: A few failure cases of the proposed method. From top to bottom, the sampled patches are from RoadTracer, DeepGlobe, Mass, DRIVE and CREMI datasets respectively.