# OpenReview forum: "Structure-Aware Image Segmentation with Homotopy Warping"
_NeurIPS.cc/2022/Conference — NeurIPS 2022 Accept_

### Official Review · Reviewer_4Rx6 · 2022-07-10

**Rating:** 7
**Confidence:** 4
**Soundness:** 3 good
**Presentation:** 3 good
**Contribution:** 3 good

**Summary:**

A topology-aware learning objective is proposed for semantic segmentation models. This objective relies upon a warping operation that transforms a predicted mask to one that is close to the ground truth mask in pixel space without changing the topology, and vice versa. The objective is a pixelwise cross-entropy computed on the points of disagreement between the target and warped source masks. Evaluation on 2D and 3D satellite and medical imagery segmentation datasets shows the algorithm is efficient and delivers strong results.

**Questions:**

- For what other kinds of data could this approach be useful? How could the proposed method be generalized to multiclass problems?
- Importance of the distance sorting heuristic: should drops in performance (accuracy or computation time) be expected if a simpler method of warping the masks is used, such as iteratively selecting a random non-simple point that borders the current mask? As another, simpler ablation study, I would suggest to evaluate warping only in one direction (ground truth to prediction or prediction to ground truth).
- L185: This line is somewhat confusing or misleading, since the proposed heuristic only finds a mask that is locally optimal, i.e., no simple points can be flipped to decrease the Hamming distance. (Is it known whether finding the global optimum simply slow (cf. L239-242) or computationally intractable?)
- L219: What would happen if each pixel's on/off state is sampled from the output distribution instead of bring thresholded?

**Limitations:**

Please see the first two points of Weaknesses above.

**Strengths And Weaknesses:**

Strengths:
- Simple, intuitive algorithm and clear writing. The algorithm is well-motivated and mathematically sound, and care is taken to make it efficient using the heuristic for computing the points of interest.
- Strong experimental evaluation on diverse datasets where preservation of connectivity structure has been an important challenge.
- Ablations and comparisons are appropriately selected.

Weaknesses:
- The paper would benefit from more discussion about what may be responsible for the better performance of the algorithm relative to baseline methods of selecting critical points. The comparison with TopoNet's points of interest in Figure 7 gives some intuition about the question, but there are no comparisons of outputs of models trained using different critical point selection methods that would show the impact of a noisy point selection on the model output.
- Similarly, I would have liked to see some examples of the failure modes of the proposed algorithm, as well as images of outputs on all the datasets studied.

---

> ### Author Response · Authors · 2022-08-02
> **Response to the comments of Reviewer 4Rx6**
>
> Thanks for your constructive feedback. Below we address specific concerns one-by-one.
>
> ---
>
> **Q1**: There are no comparisons of outputs of models trained using different critical point selection methods that would show the impact of a noisy point selection on the model output. Importance of the distance sorting heuristic: should drops in performance (accuracy or computation time) be expected if a simpler method of warping the masks is used, such as iteratively selecting a random non-simple point that borders the current mask? As another, simpler ablation study, I would suggest to evaluate warping only in one direction (ground truth to prediction or prediction to ground truth).
>
> **A1**: We conduct an additional experiment by randomly choosing a set of pixels for each iteration instead of identifying all the topologically relevant pixels.  Specifically, we randomly choose 1/10  of the whole wrongly predicted pixels as the pixel set. Here we report the performances on the RoadTracer dataset:
>
> | Method | DICE $\uparrow$ | ARI $\uparrow$ | Warping $\downarrow$ | Betti $\downarrow$ |
> |:------:|:------:|:------:|:------:|:------:|
> |  UNet | 0.587 | 0.544 | 10.412 $\times10^{-3}$ |1.591|
> | Random select | 0.584 | 0.547 | 10.261 $\times 10^{-3}$ | 1.672|
> | Warping (GT-> Pred) | 0.594 | 0.567 | 9.171 $\times 10^{-3}$ | 1.290 |
> | Warping (Pred -> GT) | 0.598 | 0.562 | 9.124 $\times 10^{-3}$ |1.315 |
> | **Warping**| **0.603** | **0.572** | **8.853** $\times 10^{−3}$ | **1.251**|
>
> As we can see from the above table, ‘Random select’ strategy achieves comparable results to the vanilla UNet, which is not surprising as the model is topology agnostic.
>
> Also, we conduct further ablation studies as suggested. The two variations (warping only in one direction: ground truth to prediction or prediction to ground truth) achieve reasonably better while still slightly inferior results to the proposed version. The reason might be that the proposed critical point selection strategy contains more complete topologically challenging locations.
> Both ablation studies demonstrate the effectiveness of the proposed critical point selection strategy.
>
> ---
>
> **Q2**: Similarly, I would have liked to see some examples of the failure modes of the proposed algorithm, as well as images of outputs on all the datasets studied.
>
> **A2**: We add a few of the failure cases in the updated version (Fig.12 in Sec.17 of the Supplementary Material) as you requested. Note that inferring topology given an image is a very difficult task, especially near challenging spots, e.g., blurred membrane locations or weak vessel connections. Current methods can help to improve the topology-wise accuracy, while they are far from perfect.
>
> ---
>
> **Q3**: For what other kinds of data could this approach be useful? How could the proposed method be generalized to multiclass problems?
>
> **A3**: In the paper we focus on the segmentation of curvilinear structures, and try to improve the topology-aware accuracy. To the best of our knowledge, we haven’t seen datasets with multiclass curvilinear structures. We note that it’s always doable to convert the multiclass segmentation task to a number of binary class segmentation tasks.
>
> ---
>
> **Q4**: L185: This line is somewhat confusing or misleading, since the proposed heuristic only finds a mask that is locally optimal, i.e., no simple points can be flipped to decrease the Hamming distance. (Is it known whether finding the global optimum simply slow (cf. L239-242) or computationally intractable?)
>
> **A4**: L185: In this part, we are talking about the possible ‘best warping’ which reaches the minimal Hamming distance. And you are right that in practice, by using the proposed algorithm, we only find a mask that is locally optimal and the ‘perfect’ minimal distance is not always guaranteed. We’ve
>
> Regarding finding the global optimum, the search space is very large. The reason is that some non-simple points might become simple as the warping algorithm continues (since their neighboring pixels get flipped in previous iterations).  Thus there are too many degrees of freedom, and finding the global optimum is computationally intractable. Moreover, we’d like to emphasize that it’s fine to miss some critical locations in one specific epoch, as we are not expecting to fix all the topological errors within one epoch.
>
> ---
>
> **Q5**: L219: What would happen if each pixel's on/off state is sampled from the output distribution instead of bring thresholded?
>
> **A5**: We can infer topology only on the binarized mask, and so we cannot directly apply warping on the output distribution (continuous likelihood map).
>
> ---

---

> > ### Comment · Reviewer_4Rx6 · 2022-08-02
> > **Clarification to questions**
> >
> > Thank you for the detailed answer. This answers most of my questions, but there are a few things I'd like to clarify, in decreasing order of importance.
> >
> > >> What would happen if each pixel's on/off state is sampled from the output distribution instead of bring thresholded?
> > >
> > > We can infer topology only on the binarized mask, and so we cannot directly apply warping on the output distribution (continuous likelihood map).
> >
> > My suggestion was to try binarizing the mask differently: not by thresholding at probability 0.5, but by sampling the binary value of each pixel from the output distribution and fixing it for the whole duration of the warping procedure. This binarization is stochastic and may vary between iterations. It seems to require just changing `torch.argmax(pre, dim=1)` to `torch.multinomial(pre, dim=1).squeeze(1)` in lines 119 and 150 of `warping_loss.py`. This is not essential, but I would like to see it if it is easy to run.
> >
> > >> Importance of the distance sorting heuristic: should drops in performance (accuracy or computation time) be expected if a simpler method of warping the masks is used, such as iteratively selecting a random non-simple point that borders the current mask?
> > >
> > > We conduct an additional experiment by randomly choosing a set of pixels for each iteration instead of identifying all the topologically relevant pixels. Specifically, we randomly choose 1/10 of the whole wrongly predicted pixels as the pixel set.
> >
> > Thanks for sharing the updated results, which are interesting and support the main claims of the paper. However, what I was proposing was to test specifically the effect of the distance heuristic: still flip only non-simple pixels -- not random wrongly predicted ones -- but removing the heuristic that uses the distance transform to choose which non-simple pixels to flip. It's also not essential, but would be interesting to see in the final version.
> >
> > >> For what other kinds of data could this approach be useful? How could the proposed method be generalized to multiclass problems?
> > >
> > > To the best of our knowledge, we haven’t seen datasets with multiclass curvilinear structures. We note that it’s always doable to convert the multiclass segmentation task to a number of binary class segmentation tasks.
> >
> > This is more of a suggestion for future work than a comment on this paper, but there are high-resolution remote sensing datasets with both roads and waterways (which can have similar appearances) and ones that separate roads from other kinds of impervious surfaces (buildings).

---

> > > ### Author Response · Authors · 2022-08-05
> > > **Further response to Reviewer 4Rx6**
> > >
> > > Thanks very much for your further clarifications! We conduct the additional ablation studies as you suggested. We'll include these additional results in the final version.
> > >
> > > ---
> > >
> > > **Q6**: My suggestion was to try binarizing the mask differently: not by thresholding at probability 0.5, but by sampling the binary value of each pixel from the output distribution and fixing it for the whole duration of the warping procedure. This binarization is stochastic and may vary between iterations.
> > >
> > > **A6**: Thanks for your clarification. It’s easy to implement and we report the performances on the RoadTracer dataset:
> > >
> > > | Method | DICE $\uparrow$ | ARI $\uparrow$ | Warping $\downarrow$ | Betti $\downarrow$ |
> > > |:------:|:------:|:------:|:------:|:------:|
> > > |   Multinomial  |  0.599 |    0.571   | 9.017 $\times 10^{-3}$| **1.245** |
> > > |  **Warping**  |  **0.603** |   **0.572**   |  **8.853** $\times 10^{-3}$| 1.251 |
> > >
> > > From the above table, we see that the ‘Multinomial’ alternative achieves very close results to the proposed method. The reason is that after going through softmax, the predicted probabilities will be close to 0 or 1 (for binary labels). As torch.multinomial returns indices based on the probability distribution, it is very similar to the proposed thresholding strategy.
> > >
> > > ---
> > >
> > > **Q7**: What I was proposing was to test specifically the effect of the distance heuristic: still flip only non-simple pixels but removing the heuristic that uses the distance transform to choose which non-simple pixels to flip.
> > >
> > > **A7**: In our understanding, you mean only flipping the simple pixels but removing the heuristic that uses the distance transform, and then use the remaining non-simple pixels as our critical pixel set. Please correct us if we misunderstood. And we report the performances on the RoadTracer dataset:
> > >
> > > | Method | DICE $\uparrow$ | ARI $\uparrow$ | Warping $\downarrow$ | Betti $\downarrow$ |
> > > |:------:|:------:|:------:|:------:|:------:|
> > > |   w/o DT  |  0.586 |    0.547   | 10.256 $\times 10^{-3}$| 1.473 |
> > > |  **Warping**  |  **0.603** |   **0.572**   |  **8.853** $\times 10^{-3}$| **1.251** |
> > >
> > > As we can see from the above table, by removing the heuristic that uses the distance transform, the performances of the method drop significantly. The reason is that without the heuristic, we only flip the wrongly predicted pixels which are near the boundaries. In this way, the extracted critical pixel set will be very noisy and most of the pixels are irrelevant to the topology.
> > >
> > > ---
> > >
> > > **Q8**: There are high-resolution remote sensing datasets with both roads and waterways (which can have similar appearances) and ones that separate roads from other kinds of impervious surfaces (buildings).
> > >
> > > **A8**: Thanks for your suggestion, and we’ll leave this as a future work.
> > >
> > > ---
> > >
> > > Please let us know if you have further questions/concerns.

---

> > > > ### Comment · Reviewer_4Rx6 · 2022-08-05
> > > > **Thank you**
> > > >
> > > > Thank you for the quick response! I have no further questions for now, and, having read all the reviews and responses, I am maintaining my positive rating of the paper.

---

> > > > > ### Author Response · Authors · 2022-08-05
> > > > > **Thank you**
> > > > >
> > > > > Thanks very much for your time!

---

### Official Review · Reviewer_RT7y · 2022-07-11

**Rating:** 5
**Confidence:** 4
**Soundness:** 3 good
**Presentation:** 2 fair
**Contribution:** 3 good

**Summary:**

The authors implement the Warping Error metric, introduced by Jain et al. (CVPR 2010) as a loss for topology aware segmentation. To be able to efficiently calculate the warping, the authors introduce a heuristic algorithm based on the distance transform to choose the pixels to "flip" or warp to achieve a better topology.

**Questions:**

How does the filtering using the DT compare to the calculation of image persistence computationally?

**Limitations:**

yes.

**Strengths And Weaknesses:**


**Strengths:**

Overall I find the proposed warping idea to be innovative and clever. It improves topology awareness in segmentation models in an efficient manner. It is significantly faster to compute than methods based on barcode-matching and persistent homology and achieves competitive performance to such methods.


**Weaknesses:**

1) Terminology: Introduction l. 24-26 "pixels near the peripheral of the object of interest can generally be challenging, but not relevant to topology." I think this statement is problematic. When considering the inverse e.g. in the case of a surface or vessel, that a foreground pixel changes to background. Such a scenario would immediately lead to a topology mismatch (Betti error 1).

2) Terminology: "topologically critical location" -->  I find this terminology to be not optimally chosen. I agree that the warping concept appears to help with identifying pixels which may close loops or fill holes. However, considering the warping I do not see a guarantee that such "locations" (as in the exact location) which I understand to refer to individual or groups of pixels are indeed part of the real foreground, nor are these locations unique. A slightly varying warping may propose a set of different pixels.

3) The identified locations are more likely to be relevant to topological errors. --> this statement should be statistically supported. Compared to what exactly? Does this rely on the point estimate for any pixel? Or given a particularly trained network?

4) Theorem 1: The presentation of a well known definition from Kong et al. is trivial and could be presented in a different way.

5) Experimentation, lack of implementation details: In Table 2 and a dedicated section, the authors show an ablation study on the influence of lamda on the results. Lamda is a linear parameter, weighting the contribution of the new loss to the overall loss. Similarly, the studied baseline methods, e.g. TopoNet [24], DMT [25], and clDice [42] have a loss weighting parameter. It would be important to understand how and if the parameters of the baselines were chosen and experimented with. (I understand that the authors cannot train ablation studies for all baselines etc.) However, it is an important information to understand the results in Table 1.

6) Terminology: l. 34 "to force the neural network to memorize them" --> I would tone down this statement, in my understanding, the neural network does not memorize an exact "critical point" as such in TopoNet [24].



**Minor:**

- I find the method section to be a bit wordy, it could be compressed on the essential definitions.

- There exist several grammatical errors, please double-check these with a focus on plurals and articles. E.g. l. 271 "This lemma is naturally generalized to 3D case."

- l. 52 "language of topology" I find this to be an imprecise definition or formulation.

**Note:**

After rebuttal and discussion I increased the rating to 5.

---

> ### Author Response · Authors · 2022-08-02
> **Response to the comments of Reviewer RT7y**
>
> Thanks for your constructive feedback. We will improve our manuscript accordingly, especially for the presentation issues. Below we address specific concerns one-by-one.
>
> ---
>
> **Q1**: Terminology
>
> **A1**: Thanks for pointing out all of them.
>
> * I.24-26: We change it to "pixels at the boundary of the object of interest can generally be challenging, but not relevant to topology".
>
> * "topologically critical location": Yes, you are right. Based on our algorithm, the identified locations are not unique. We use the term ‘topologically critical location’ for simple understanding of the paper.
>
> *  l. 34 "to force the neural network to memorize them": Thanks for your suggestions. We changed "to force the neural network to memorize them" to "to force the neural network to focus on them".
>
> ---
>
> **Q2**: The identified locations are more likely to be relevant to topological errors. --> this statement should be statistically supported. Compared to what exactly? Does this rely on the point estimate for any pixel? Or given a particularly trained network?
>
> **A2**: It is compared to the whole set of wrongly predicted locations. We mean that not all the wrongly predicted locations are related to topology, and thus we use the proposed warping strategy to extract the topologically relevant locations. The identified locations are independent of the network, and are obtained  from the definition of non-simple points.
>
> ---
>
> **Q3**: Theorem 1: The presentation of a well known definition from Kong et al. is trivial and could be presented in a different way.
>
> **A3**: Thanks for your suggestion. We changed ‘Theorem 1’ to ‘Definition 1’, which is referred to as the definition of simple points in the 2D case.  We wanted to highlight this definition, as it is essential for the understanding of our warping strategy/loss.
>
> ---
>
> **Q4**: It would be important to understand how and if the parameters of the baselines were chosen and experimented with. (I understand that the authors cannot train ablation studies for all baselines etc.) However, it is an important information to understand the results in Table 1.
>
> **A4**: This is a very good question.
>
> For the Mass, DRIVE and CREMI dataset, we follow the settings in DMT [26]. Specifically, we generate the numbers of ‘Warping Error’ for all the methods, and for the remaining metrics we copy the numbers from the original paper. We believe the parameters have been fine-tuned and carefully chosen.
>
> For the RoadTracer and DeepGlobe datasets, we tune the parameters of the baselines (though it may not be exhaustive) as well as the proposed method, and report the best performances with the appropriate parameters.
>
> ---
>
> **Q5**: How does the filtering using the DT compare to the calculation of image persistence computationally?
>
> **A5**: We’ve compared the efficiency of different methods in Tab.4 of the main paper, including the algorithm complexity and training time in practice. We discuss the same in L389.
>
> ---

---

> > ### Author Response · Authors · 2022-08-07
> > **Follow up**
> >
> > Dear Review RT7y,
> >
> > Will you be able to kindly spend some time to have a look at our response?
> >
> > We'll try to resolve your further questions/concerns (if any) during the discussion period.
> >
> > Best,
> >
> > Authors of Paper #11428

---

> > ### Comment · Reviewer_RT7y · 2022-08-07
> > **Discussion of Rebuttal**
> >
> > Dear authors,
> >
> > thank you for replying to my review and considering my comments. Many of my quesitons are addressed and I am willing to increase the rating. Please see additional comments here:
> >
> > 1) (Review Q 2) I am still unsure about the terminology, to me a "topologically critical location" would be a unique pixel location. Maybe the authors are able to come up with a better terminology along the lines of warping.
> >
> > 2) (Review Q 5)  Experimentation and reproducibility: the authors state in their checklist that they provide all details, data, code and hyperparameters to reproduce the experimental results (checklist 3 a, b). To me the current supplementary material does not specify all of these. E.g. which images are in the train/test/val sets and which parameters were chosen for which ablation study in order to reproduce the numbers in the experimentation. I would encourage the authors to release all of the above in a github repository.

---

> > > ### Author Response · Authors · 2022-08-07
> > > **Thanks for your response**
> > >
> > > Dear Reviewer RT7y,
> > >
> > > Thanks very much for your time and additional comments! Below we are trying to resolve your further questions/concerns.
> > >
> > > > I am still unsure about the terminology, to me a "topologically critical location" would be a unique pixel location. Maybe the authors are able to come up with a better terminology along the lines of warping.
> > >
> > > Thanks for your suggestion. In the paper, _topologically critical locations_ refer to the set of identified critical points/locations which are relevant to topology. Will _topologically critical pixel set_ a better term for you?
> > >
> > > > Experimentation and reproducibility.
> > >
> > > As we have mentioned in the Sec.11 (Datasets section) of the updated version, for the RoadTracer and DeepGlobe datasets, we follow the data splits of [4] and [5], respectively. And for the other datasets, such as Massachusetts, DRIVE and CREMI, we conduct a three-cross validation, which follows the settings in DMT [26].
> > >
> > > And in Tab.7, we provide the loss weights $\lambda$ (hyperparameter) for each dataset.
> > >
> > > But as you suggested, we'll provide all the necessary details in a public github repo upon acceptance.

---

> > > > ### Comment · Reviewer_RT7y · 2022-08-08
> > > > **Reviewer Interaction 2**
> > > >
> > > > Dear authors,
> > > > I do think that the term "topologically critical pixel" is a better fit for the given method. I have now increased my rating.

---

> > > > > ### Author Response · Authors · 2022-08-08
> > > > > **Thanks very much for your positive feedback**
> > > > >
> > > > > Dear Reviewer RT7y,
> > > > >
> > > > > Thanks very much for your positive feedback! We'll make the modifications accordingly in the final version.
> > > > >
> > > > > Best,
> > > > >
> > > > > Authors of Paper #11428

---

### Official Review · Reviewer_6qMj · 2022-07-11

**Rating:** 6
**Confidence:** 4
**Soundness:** 3 good
**Presentation:** 3 good
**Contribution:** 3 good

**Summary:**

This paper presents a new method to improve the topology of image segmentation.  The method finds topologically critical locations by comparing the predicted segmentation mask with the ground truth, and proposes a novel homotopy warping loss which guides the training to improve topological accuracy. Experiments are conducted in both 2D and 3D settings to segment satellite and biomedical images, and results show the proposed method outperforms other existing methods.

**Questions:**

1.	All the used cases in the paper address the segmentation error in between the segmented areas. What if some False Positives or False Negatives exist at the edges of segmentation regions?

2.	The datasets used in the proposed study have straight or curved lines as an area of interest. Would the method be useful for datasets with more irregular shapes as a region of interest—for example, the Kvasir SEG - Simula dataset?  https://arxiv.org/abs/1911.07069

3.	How might it be possible to apply the method for image segmentation problems with more than two classes (e.g. a problem with two foreground classes and one background)?

4.	What are the limitations of the proposed method?


**Limitations:**

Limitations are not discussed in the paper.

**Strengths And Weaknesses:**

Strengths:

1.	Although previous work such as TopoNet also emphasize the importance of focussing on topologically critical locations, the warping approach taken in this paper appears to be novel and more efficient, computationally.

2.	The warping loss is intuitive and seeks to transform the topology of the predicted segmentation mask to match the ground truth (and vice versa).  The proposed algorithm focusses on simple points, which are described clearly in terms of their connectivity.

3.	The experimental results demonstrate the method is capable of improving segmentation not just at the critical points but overall in the image.

Weaknesses:

1.	The segmentation performance obtained from this method mainly relies on the standard UNet results. If the segmentation output from UNet is noisy, the warping operation may take many iterations, perhaps more so initially as the network is starting to train.  Experiments with other architectures is left for future work.

2.	While results are improved, the improvement may be subtle; for example 1% improvement in Dice score for most experiments in Table 1.  Nonetheless, visual results in Figure 6 show good improvement.

3.	It wasn’t clear to this reviewer if or how the method generalises to problems where there are more segmentation classes (e.g. three or more).  All the experiments shown in the paper are binary segmentation problems.  However, the paper makes it clear that it is focussing on binary segmentation.

---

> ### Author Response · Authors · 2022-08-02
> **Response to the comments of Reviewer 6qMj**
>
> Thanks for your constructive feedback. Below we address specific concerns one-by-one.
>
> ---
>
> **Q1**: The segmentation performance obtained from this method mainly relies on the standard UNet results. If the segmentation output from UNet is noisy, the warping operation may take many iterations, perhaps more so initially as the network is starting to train.
>
> **A1**: Our basic motivation is to identify the topologically challenging locations and then force the neural networks to focus on/correct them. In practice, the output of the standard UNet is usually reasonable (since it’s optimized with per-pixel loss), and so the number of epochs we observed was decent. Also, the warping is operated on the binarized mask instead of the noisy likelihood map. In this way, the proposed method can always efficiently identify and focus on the topologically challenging locations.
>
> ---
>
> **Q2**: Experiments with other architectures is left for future work.
>
> **A2**: We add experiments by using FCN as the backbone, and we report the performances on the RoadTracer dataset.
>
> | Method | DICE $\uparrow$ | ARI $\uparrow$ | Warping $\downarrow$ | Betti $\downarrow$ |
> |:------:|:------:|:------:|:------:|:------:|
> |   FCN  |  0.567  |    0.518   | 14.515 $\times 10^{-3}$| 2.311 |
> |  FCN + Warping  |  **0.581** |   **0.543**   |  **11.126** $\times 10^{-3}$| **1.918** |
>
> From the results, we observe that the proposed method also improves the topology-aware segmentation accuracy, regardless of the network backbones.
>
> ---
>
> **Q3**: While results are improved, the improvement may be subtle; for example 1% improvement in Dice score for most experiments in Table 1.
>
> **A3**: In Sec.10 of the original version (which corresponds to Sec.15 in the revised version) and Tab.6 of the supplementary material, we provide stddev besides mean, and use t-test (95% confidence interval) to determine the statistical significance (highlighted with bold) for the RoadTracer dataset. The quantitative results show that the proposed method performs significantly better than baselines. Also, we believe that the topology-relevant metrics, such as ARI, Warping Error and Betti Error are better metrics for topology/structure-aware image segmentation tasks.
>
> ---
>
> **Q4**: It wasn’t clear to this reviewer if or how the method generalizes to problems where there are more segmentation classes (e.g. three or more).
>
> **A4**: In the paper, we mainly focus on curvilinear structure segmentation tasks, and try to improve the topology-aware accuracy. To the best of our knowledge, we haven’t seen datasets with multiclass curvilinear structures. Additionally it’s always doable to convert the multiclass segmentation task to a number of binary class segmentation tasks.
>
> ---
>
> **Q5**: What if some False Positives or False Negatives exist at the edges of segmentation regions?
>
> **A5**: The proposed method mainly focuses on the topological errors of curvilinear structures. The logic is that the standard segmentation methods (such as vanilla UNet optimized with dice loss) can deal with most of the pixel level segmentation. If False Positives/Negatives exist at the edges of segmentation regions, they won’t be identified as topologically challenging locations and corrected  by the proposed method, and actually they will not induce topological errors.
>
> ---
>
> **Q6**: Would the method be useful for datasets with more irregular shapes as a region of interest—for example, the Kvasir SEG - Simula dataset?
>
> **A6**: In this paper we mainly test our method on fine-scale curvilinear structure datasets. However we believe it could be extended to more general cases. As for the Kvasir SEG - Simula dataset you mentioned, the proposed method should help to improve the topology-wise accuracy. The reason is that the proposed method can always help to capture the topological errors (if any) between the prediction and ground truth masks, and force the neural networks to focus on/correct these topological errors. For this specific dataset, if the predicted mask has a different number of connected components compared to the ground truth mask (which means topological errors), our proposed method will identify those pixels which induce the topological errors and force the neural network to focus on and correct them.
>
> ---
>
> **Q7**: What are the limitations of the proposed method?
>
> **A7**: One major limitation might be that currently we only test the proposed method on curvilinear structure datasets, and we’ll try to explore more general cases in the future.
>
> ---

---

> > ### Comment · Reviewer_6qMj · 2022-08-08
> > **Response**
> >
> > Thank you authors for the response.  I have no further questions and will maintain my postive rating.  Should the paper get accepted it would be helpful to comment a bit more on the limitations of the method as this may seed future research.

---

> > > ### Author Response · Authors · 2022-08-08
> > > **Thanks for your response**
> > >
> > > Dear Reviewer 6qMj,
> > >
> > > Thanks very much for your confirmation and suggestions! We'll include a separate limitation section/paragraph in the final version as you suggested.
> > >
> > > Best,
> > >
> > > Authors of Paper #11428

---

### Official Review · Reviewer_xhKW · 2022-07-15

**Rating:** 6
**Confidence:** 3
**Soundness:** 4 excellent
**Presentation:** 4 excellent
**Contribution:** 3 good

**Summary:**

This paper proposed a new method for segmenting images with better topological accuracy. Specifically, a new algorithm utilizing distance transform is used to identify the topologically critical locations, based on which a new homotopy warping loss is used to measure the difference between the predicted and ground-truth topology and train the segmentation network. Experiments on four 2D datasets and one 3D dataset validated the efficacy of the proposed method. Besides pixel-wise accuracy, better topology accuracy is obtained.

**Questions:**

Please see the comments above

**Strengths And Weaknesses:**

Strengths
1. This paper introduced an efficient heuristic algorithm to identify topologically critical locations. Compared with a previous work on topology-preserving image segmentation method which runs in cubic time complexity, the proposed algorithm's complexity is linear to image size, thus making the training more efficient.

2. A new homotopy warping loss is proposed to train the segmentation network to preserve the topology. The homotopy warping loss penalizes errors on topologically critical locations and results in models with better topological accuracies.

3. Comparisons with existing topology-preserving segmentation methods show that the proposed method achieves the best performance. Ablation studies provide some guidance regarding the choices of loss weights and loss functions.

Weaknesses
1. The proposed distance-ordered homotopy warping algorithm is possible to miss some simple pixels during the flipping although very rare.

2. In the previous work TopoNet, Variation of Information (VOI) is used as a metric to measure topological correctness. It would be better if VOI is also compared in the experiments.

3. Grammar/spelling errors:
a) Line 92: "Except for the methods mentioned above, UNet has also been one of the most popular methods for image segmentation". It seems "besides" is the proper preposition, not "except for".
b) Line 131: "efficiently identity" -> "efficiently identify"
c) Line 382: "we'd like the investigate" -> "we investigate"

---

> ### Author Response · Authors · 2022-08-02
> **Response to the comments of Reviewer xhKW**
>
> Thanks for your constructive feedback. Below we address specific concerns one-by-one.
>
> ---
>
> **Q1**: The proposed distance-ordered homotopy warping algorithm is possible to miss some simple pixels during the flipping although very rare.
>
> **A1**: Yes, and we had already mentioned this in L277-282. Moreover, we’d like to emphasize that it’s fine to miss some simple points during flipping in one specific epoch, as we are not expecting to fix all the topological errors within one epoch.
>
> ---
>
> **Q2**: In the previous work TopoNet, Variation of Information (VOI) is used as a metric to measure topological correctness. It would be better if VOI is also compared in the experiments.
>
> **A2**: We omitted the VOI in the original version because of space limitation, and we report the VOIs of the RoadTracer dataset here.
>
> | Method | VOI $\downarrow$|
> |:------:|:------:|
> |   UNet  |  2.318  |
> |  RoadTracer  |  2.542 |
> |    VecRoad  |    2.419 |
> |    iCurb  |    2.251 |
> |    VGG-UNet  |   2.109 |
> |    TopoNet  |    2.088 |
> |    clDice  |    2.185 |
> |    DMT  |    1.967 |
> |    **Warping**  |    **1.896** |
>
> From the table, we see that the proposed method also achieves the best VOI compared with the other baseline methods. We’ll include the VOIs for the other datasets in the final version.
>
> ---
>
> **Q3**: Presentation issues.
>
> **A3**: Thanks for your suggestions. We’ve fixed the presentation issues in the revised version.
>
> ---

---

> > ### Comment · Reviewer_xhKW · 2022-08-08
> > **Response to authors' update**
> >
> > Thank you for answering my questions. I have no further questions and will maintain my rating.

---

> > > ### Author Response · Authors · 2022-08-08
> > > **Thanks for your response**
> > >
> > > Dear Reviewer xhKW,
> > >
> > > Thanks very much for your confirmation!

---

### Author Response · Authors · 2022-08-02
**General response**

We thank all the reviewers for their valuable feedback! We will improve our presentations accordingly. We are encouraged that all reviewers appreciated the novelty of the contribution and the performances on challenging benchmarks. We have updated the revised version and added the supplementary material (which was a separate pdf file for the original submission) at the end of the main paper for friendly reading.

Below we address specific concerns one-by-one.

---

### Meta-Review · Area_Chair_wpaM · 2022-08-23

**Recommendation:** Accept
**Confidence:** Certain

**Metareview:**

The paper proposes a topology-aware learning objective for semantic segmentation models based upon warping masks. The loss is used for training satellite data and medical segmentation datasets and provides benefits in these domains. Reviewers acknowledge that the approach is simple, intuitive, has thorough empirical evaluation and ablations. Reviewers note a few presentation issues, e.g. related to terminology. These must be fixed in a final revision of the paper. Overall all reviewers vote for acceptance and so do I.

**Award:**

No

---

### Decision · Program_Chairs · 2022-09-14

Accept